# Effect of the Intake of a Snack Containing Dietary Fiber on Postprandial Glucose Levels

**DOI:** 10.3390/foods9101500

**Published:** 2020-10-20

**Authors:** Hyeon-Ki Kim, Takuya Nanba, Mamiho Ozaki, Hanako Chijiki, Masaki Takahashi, Mayuko Fukazawa, Jin Okubo, Shigenobu Shibata

**Affiliations:** 1Faculty of Science and Engineering, Waseda University, 2-2 Wakamatsu-cho, Shinjuku, Tokyo 1628480, Japan; hk.kim@aoni.waseda.jp; 2Graduate School of Advanced Science and Engineering, Waseda University, 2-2 Wakamatsu-cho Shinjuku, Tokyo 1628480, Japan; n-x.t.x-n@asagi.waseda.jp (T.N.); mo_u2@ruri.waseda.jp (M.O.); hnk-1022@akane.waseda.jp (H.C.); yu8m1omo5iwo6@suou.waseda.jp (M.F.); zin_eile_0kb@akane.waseda.jp (J.O.); 3Institute for Liberal Arts, Tokyo Institute of Technology, 2-12-1, Ookayama Meguro-ku, Tokyo 152-8550, Japan; takahashi.m.bp@m.titech.ac.jp

**Keywords:** snack, dietary fiber, postprandial glucose, older adult

## Abstract

To examine the effects of the intake of a snack containing dietary fiber under free-living conditions on postprandial glucose levels in older adults, nine healthy older adults aged 76.9 ± 1.6 years (mean ± standard error) completed two crossover trials: 1) regular snack (BISCUIT) intake and 2) intake of snacks with a high dietary fiber content (DF-BISCUIT). In both trials, each participant consumed either BISCUIT or DF-BISCUIT between lunch and dinner time for 1 week. During the intervention, the blood glucose levels of all the subjects were observed using a continuous glucose monitoring system. Lower 24 h blood glucose levels were yielded in the DF-BISCUIT than the BISCUIT trials. Moreover, compared to the BISCUIT trials, the blood glucose levels after dinner and areas under the curve (AUCs) were significantly decreased in the DF-BISCUIT treatments. The blood glucose levels and AUCs after the intake of the next day’s breakfast were suppressed in the DF-BISCUIT treatments compared to those in the BISCUIT trials. Our data indicate that the intake of snacks with a high dietary fiber content under free-living conditions is an effective way to restrain postprandial glucose levels and that the effect lasts until breakfast the next day.

## 1. Introduction

Elevated postprandial glucose level is associated with the risk of type 2 diabetes and cardiovascular disease [1,2,3]. Previous studies have reported that glucose metabolism is effected not only by the size of meals but also by the timing of their intake [4,5]. In fact, even in the case of a similar meal, at dinner, it is observed that the blood glucose concentration is higher than that at breakfast and the secretion levels of insulin and the hormones that promote its secretion are reduced [6,7,8]. That is, there is a potential that during dinner the glucose metabolism function degree decreases and blood glucose levels increase. Therefore, the control of blood glucose levels after dinner is important for the prevention of diabetes and cardiovascular disease and for the improvement of the disease state.

The consumption of snacks contributes substantially to a person’s daily energy intake. A recent study reported an increase in the energy intake level from snacks consumed between lunch and dinner [9]. Furthermore, it has been shown that more than half of all people with type 2 diabetes consume snacks at least twice a week [10]. The ingestion of snacks may increase the risk of diabetes and obesity due to elevated blood glucose levels [11,12]. It has been shown that the intake of proper snacks suppresses a rise in blood glucose levels due to a subsequent meal. A previous study demonstrated that the intake of snacks at an adequate interval from lunch is an effective way to control postprandial glucose levels and glycemic excursions at dinner [13]. Therefore, proper snacking may be useful for the control of postprandial blood glucose levels. In addition, differences in the snack components may play an important role in glucose metabolism regulation and contribute to the antidiabetic effect.

Dietary fiber plays a role in the attenuation of postprandial glycemia [14]. In particular, the viscosity of dietary fiber has been shown to affect the physiological response to glucose absorption [15,16]. Viscous dietary fibers become thicker when mixed with a liquid. Such characteristics of dietary fiber are associated with prolonged starch digestion and absorption rates, resulting in altered blood glucose levels [15,16]. Therefore, consumption of a snack that contains dietary fiber may affect the blood glucose response to a subsequent meal. However, the effects of different snack dietary fiber components on postprandial blood glucose levels remain unclear.

In addition, although previous studies have shown that snack intake has a positive impact on postprandial glucose levels [13,17], research focusing on the consecutive intake (i.e., 1 week) of snacks under free-living conditions is insufficient. Given the increasing rate of snacking in daily life [9], it is important to consider the establishment of more effective snacking guidelines for improved health. Therefore, in this study, we purposed to investigate the effect of the intake of a snack that contains dietary fiber under free-living conditions on postprandial glucose levels excursions, and we hypothesized that the consumption of a snack containing dietary fiber at late afternoon is effective in suppressing the subsequent increase in postprandial glucose levels at dinner.

## 2. Materials and Methods

### 2.1. Study Participants

Nine older adults (70–85 years old; four men and five women) participated in this study after providing written informed consent. This study was approved by the Ethics Committee of Waseda University (approval no. 2018-074) and was conducted according to the guidelines established in the Declaration of Helsinki. The human trial of the present study is registered at www.umin.ac.jp/ctr/ as UMIN000033480. Participants were recruited only if they met the following criteria: (1) not using of glucose-/insulin-lowering or related medications, (2) lack of blood pressure control (systolic blood pressure >140 mmHg and diastolic blood pressure <90 mmHg), (3) not diagnosed with dyslipidemia or diabetes by a doctor, and not taking any anti-obesity, anti-oxidant, or anti-diabetes supplement. In this study, all participants who met the inclusion criteria completed all trials.

### 2.2. Snack Contents

Ezaki Glico Co provided both the regular snack (BISCUIT) and snack with a high dietary fiber content (DF-BISCUIT) used in the present study. As summarized in Table 1 DF-BISCUIT contained 18.4 g/day total carbohydrate (9.2 g sugar and 9.2 g dietary fiber), while BISCUIT contained 18.5 g total carbohydrate (18.1 g sugar and 0.4 g dietary fiber). Each snack was also matched and adjusted based on appearance and flavor, such that they could not be distinguished.

### 2.3. Main Trials

A randomized cross-over design was used. Each participant underwent two trials in a randomized order: (1) the DF-BISCUIT treatment and (2) the BISCUIT trial (Figure 1). In both trials, participants were required to consume a snack between lunch and dinner for one week. All participants were asked to visit the laboratory for anthropometry measurements after a 12 h overnight fast.

Anthropometric variables were measured at the baseline and after two weeks. The body height was measured to the nearest 0.1 cm using a wall-mounted stadiometer (seca213, As One Corporation, Japan). Body weight was measured to the nearest 0.1 kg using a digital scale (InBody 270, InBody Co., Ltd., Tokyo, Japan). The body mass index (BMI) was calculated using the weight in kilogram divided by the square of height in metres.

### 2.4. Blood Glucose Level and Analysis

Participants were asked to wear a continuous glucose monitoring system (FreeStyle Libre Pro Blood Glucose Monitoring System) for the continuous mensuration of blood glucose levels during the study. The blood glucose level and area under the curve (AUC) 4 h after the consumption of each meal and 2 h after the snack consumption were calculated in both trials. Also, the parameters for evaluating glycemic variability were: the mean amplitude of glycemic excursion (MAGE), minimum glucose level (MIN), maximum glucose level (MAX), standard deviation (SD) of glucose level, and coefficient of variation (CV). SD and CV are two statistical values that provide a different merkmal to look at glucose variability. SD is a measure of the spread in glucose readings around the average. However, since SD needs to be interpreted in consideration of the mean blood glucose level, the CV will be used. SD is strongly influenced by the mean glucose level, while CV helps to “correct” and normalize glucose variability. The SD and MAGE values were calculated from 1200 h to 1200 h in next noon, as described previously [13], and MAX and MIN were highest and lowest glucose values, respectively.

### 2.5. Standardization of the Meal and Physical Activity

When the participants visited the laboratory at the baseline (before intervention) and after the intervention measurement, they were asked to refrain from eating breakfast for the minimization of the influence on body composition by meal intake. All participants were instructed to maintain their usual daily dietary patterns during the experiment period. Also, all participants were required to abstain from remaining inactive or participating in strenuous physical exercise throughout the entire study period. Daily energy intake and dietary fiber intake were determined from the food frequency questionnaire completed by the participants at the baseline and after the intervention. The average energy intake was labeled as kcal/d, whereas daily dietary fiber intake was shown as grams per day (g/d).

Additionally, in order to determine the daily physical activity levels (moderate-to-vigorous physical activity [MVPA], step counts), all participants were asked to wear a triaxial accelerometer (Active style Pro HJA-750C, Omron Co., Ltd., Kyoto, Japan). The participants were asked to wear the accelerometer every day at all times from the time of rising in the morning until bedtime in the evening, except during shower times. The data were only considered valid if the participants wore the accelerometer for at least a total of 10 h (600 min) daily for at least 2 weekdays and one weekend day [18]. All values ≥3 metabolic equivalents were classified as MVPA.

### 2.6. Statistical Analysis

Data analysis was performed using the Predictive Analytics Software for Windows (SPSS Japan Inc., Tokyo, Japan). All parameters were tested for normal distribution using the Kolmogorov–Smirnov test. In order to compare the postprandial glucose level and changes in the diurnal blood glucose level between the trials (DF-BISCUIT or BISCUIT), a two-factor analysis of variance (ANOVA) was used to determine the effects of the trial and time as factors. When there is a significant main or interaction observed, Bonferroni method was used for post-hoc comparisons. Pearson’s correlation coefficient was used to evaluate the association between peak glucose levels after breakfast and dinner in both trials. To investigate the change in the blood glucose levels from the baseline to the after-experiment, we used a paired *t*-test after the confirmation of the normality of all data. Results with *p*-values of less than 0.05 were considered significant.

## 3. Results

The participants’ physical characteristics between the baseline and after 2 weeks did not differ significantly (Table 2). Also, there were no significant differences in the MVPA, the number of step counts, and mealtime between the DF-BISCUIT treatments and BISCUIT trials (Table 1). The SD values in the DF-BISCUIT treatments were significantly lower than those in the BISCUIT trials, whereas there were no significant differences in the CV, MAX, MIN, and MAGE between the trials (Table 3).

The average blood glucose levels were lowered in the DF-BISCUIT treatments compared to the BISCUIT trials; however, the difference was not significant (Figure 2a). Also, the AUC between the trials did not differ significantly (Figure 2b).

We compared the blood glucose level 2 h after snack intake. For fluctuations in the blood glucose levels after the consumption of snacks and dinner, trial × time interactions (*p* = 0.001, *p* = 0.019) were observed. In the DF-BISCUIT treatments, the blood glucose levels at 90 (*p* = 0.024) and 105 min (*p* = 0.017) after snack intake were significantly lower than those in the BISCUIT trial (Figure 3c). Additionally, in the DF-BISCUIT treatments, the blood glucose levels at 15 (*p* = 0.017), 30 (*p* = 0.002), 45 (*p* = 0.004), 60 (*p* = 0.022), 120 (*p* = 0.011), and 135 min (*p* = 0.038) after dinner were significantly lower than those in the BISCUIT trial (Figure 3e). The AUCs after dinner were significantly more decreased in the DF-BISCUIT treatments than those in the BISCUIT trials (*p* = 0.002, Figure 3f). In this study, we compared the effects of snack intake on blood glucose fluctuations between sexes. Males showed similar variations to the overall results, but females did show small differences between DF-BISCUIT treatment and BISCUIT trial. Therefore, the effect of snack consumption rich in dietary fiber on blood glucose fluctuations may differ depending on the sex. However, due to the small number of participants, statistical processing is not performed. For fluctuations in the blood glucose levels after breakfast and lunch intake, trial × time interaction was not observed. However, in the DF-BISCUIT treatments, the blood glucose levels after the consumption of the next day’s breakfast were suppressed, and a significant decrease was observed compared to the BISCUIT trials in terms of the AUC (Figure 3h). No significant difference in the blood glucose levels was observed after lunch in both trials (Figure 3a,b).

In both trials, the peak blood glucose levels after each meal were examined. After dinner and after breakfast on the following day, the DF-BISCUIT treatment showed significantly lower levels than the BISCUIT trial (Figure 4c,d). However, there was no significant difference between the trials after the intakes of snacks and lunch. Furthermore, a significant correlation was observed between the peak glucose level at breakfast and peak glucose level at dinner in the DF-BISCUIT treatment (r = 0.693; *p* = 0.038) (Figure 4f).

## 4. Discussion

The main results of the present study are that the intake of a snack containing dietary fiber attenuated blood glucose levels after dinner as well as after breakfast the next day. To the best of our knowledge, the present study is the first to investigate the effects of the consumption of a snack that contains dietary fiber on postprandial glucose levels.

Dietary fiber plays a role in reducing the degree of postprandial glucose elevations [14]. Previous studies have shown that adequate dietary fiber intake is significantly associated with a reduced risk of death from diabetes and cardiovascular disease [19,20]. In the present study, the blood glucose levels were lower in the DF-BISCUIT treatments than the BISCUIT trials. These effects may be related to the physicochemical properties of dietary fiber [15]. The consumption of dietary fiber can delay the rate of gastric emptying due to the increased viscosity of gastric contents, which in the small intestine serves as a barrier to starch and oligosaccharide access by digestive enzymes. The viscosity of dietary fiber also acts as a physical barrier to slow the rate of the absorption of glucose molecules into the intestine. The ingestion of sufficient levels of dietary fiber by these mechanisms slows down the rate of glucose absorption and improves insulin function.

In addition, the intake of snacks with a high dietary fiber content between lunch and dinner led to a lower postprandial glucose response level after dinner and after breakfast the next day. This result can be explained by the second-meal phenomenon [21]. Previous studies have reported that the initial glucose load affects the glycemic response of the glucose load taken within 12 h [22,23]. In addition to the amount of glucose, its bioavailability also affects the glucose tolerance in the next meal [24,25]. In particular, in the case of a diet containing dietary fiber, the response to a subsequent dietary load is reduced. The mechanism of this phenomenon remains unknown but has been explained by biochemical or physiological factors such as free fatty acids, insulin responsiveness, gastric emptying, and absorption [26,27,28,29].

The effect of dietary fiber intake on glucose metabolism depends on the type of dietary fiber. The types of dietary fiber used in this study include isomaltodextrin (IMD), inulin, cellulose, and resistant starch. IMD is a novel highly branched α-glucan and is expected to function as a water-soluble dietary fiber [30]. A previous study showed that IMD attenuates postprandial blood glucose levels [30]. One possible mechanism is that IMD may reduce the magnitude of postprandial blood glucose excursions after carbohydrate ingestion by the inhibition of glucose absorption and enzyme activity [30]. Inulin is a water-soluble dietary fiber and is found abundantly in foods such as onions and Jerusalem artichoke. It provides several nutritional and health benefits to humans [31]. Previous studies have shown that inulin reaches the large intestine without being degraded by enzymes and is fermented by intestinal microorganisms for the improvement of the intestinal environment [32,33]. A previous study has shown that a 7-day inulin intake changes the intestinal environment in humans [34]. Therefore, even a week’s intake may be sufficiently adapted. The intestinal environment is related to insulin sensitivity and glucose metabolism, and it has been reported that increasing the level of good bacteria is effective for blood glucose control [35]. Some studies have evaluated the effect of cellulose, an insoluble dietary fiber, on blood glucose levels in humans [36], reporting a decrease in postprandial glucose levels by its intake [37,38]. In other previous studies, the effect of consumption of resistant starch in blood glucose and insulin sensitivity has been reported, which indicated that resistant starch intake improves fasting blood glucose levels and insulin sensitivity [39,40]. Insoluble dietary fiber is involved in postprandial blood glucose changes by the acceleration of the secretion of glucose-dependent insulinotropic polypeptide (GIP) [36]. GIP is an incretin hormone that stimulates postprandial insulin secretion. In this study, the ingestion of dietary fiber-containing snacks had a beneficial effect not only on subsequent postprandial glucose changes but also on the postprandial glucose changes on the following day. These results indicate that, in addition to the effect of the viscosity of the dietary fiber, a synergistic effect of the characteristics of each dietary fiber was present, and this effect may have been maintained until the next day. However, to confirm whether or not a synergistic effect was present, it is necessary to conduct research on the type of dietary fiber included in snacks. Furthermore, since there is different sugar contents in the snacks used in both trials in this study, this effect is undeniable. Therefore, it will be necessary to consider the difference in dietary fiber intake after adjusting the amount of sugar. In contrast, the total energy content of the snacks in both trials is similar. Therefore, it is considered that the influence of both trials on the blood glucose level fluctuation is largely due to the difference in dietary fiber contained in the snack.

There were several limitations to our study. First, the participants with hyperglycemia or diabetes were not included in this study. Thus, it will be important to investigate the effects of the intake of a snack containing dietary fiber on postprandial glucose levels in healthy younger adults, or patients with diabetes. Second, the meals did not provide during the entire study. Thus, differences in daily dietary intake may be influenced by individual postprandial blood glucose levels. However, participants were asked to not change their usual lifestyle such as dietary intake, sleep/wake cycle, and physical activity during the experimental period. Actually, total energy intake before and after intervention did not differ significantly. Therefore, the effect of differences in meal intake on our findings is likely to be small. It will be required to conduct studies that provide the test meal to all subjects during the intervention period. Third, the carryover effect in this study cannot be denied. Therefore, to minimize the carryover effect, the number of participants who started each trial was divided equally as much as possible. Thus, the carryover effect between both trials is considered to be small. Finally, the sample size in the current study is too small to conclude about the effects of the intake of a snack containing dietary fiber on postprandial glucose levels. This effect should be investigated in future research with a larger number of test subjects and also with the point of sex differences.

## 5. Conclusions

Our data demonstrate that the intake of dietary fiber-rich snacks under free-living conditions is an effective way to restrain postprandial glucose levels. Furthermore, it is suggested that the effect lasts until breakfast the next day.

## Figures and Tables

**Figure 1 foods-09-01500-f001:**
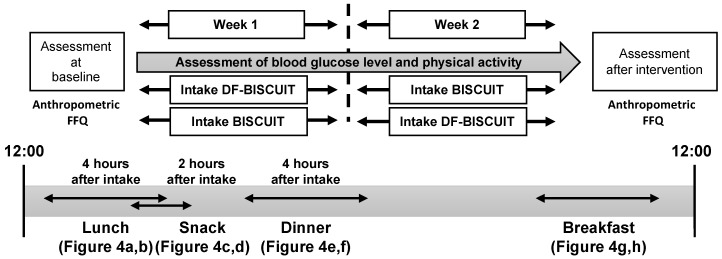
Study protocol. FFQ, food frequency questionnaire.

**Figure 2 foods-09-01500-f002:**
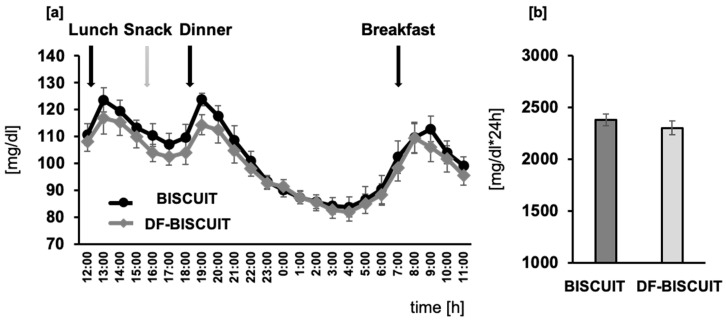
24-h fluctuations in the blood glucose levels (**a**) and areas under the curve (AUCs) (**b**).

**Figure 3 foods-09-01500-f003:**
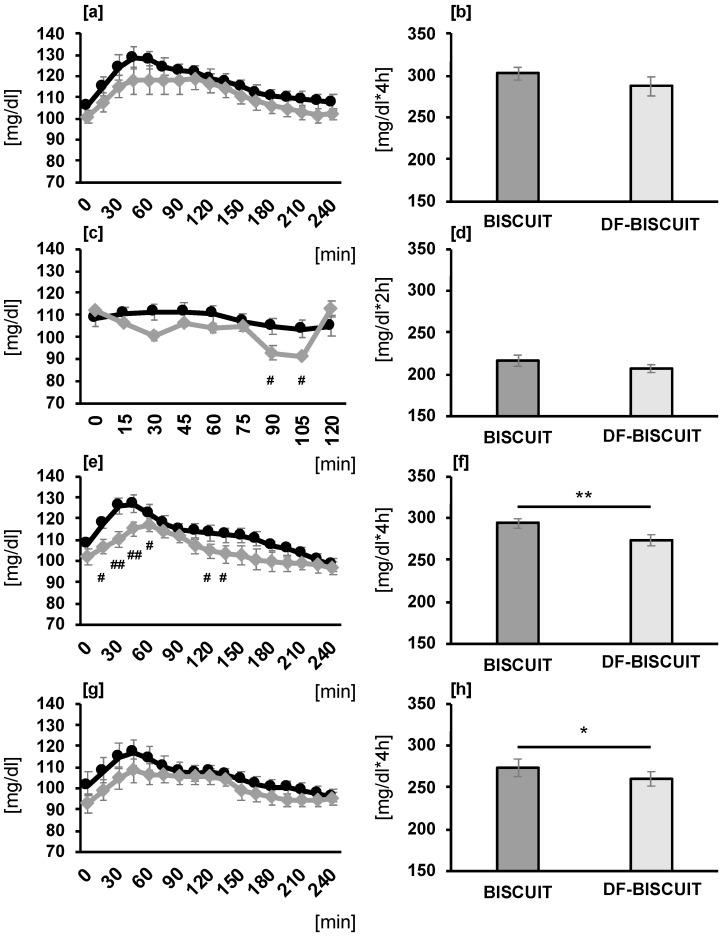
Concentrations of glucose and areas under the curve (AUCs) of lunch (**a**,**b**), snacks (**c**,**d**), dinner (**e**,**f**), and breakfast (**g**,**h**) in the BISCUIT trial and DF-BISCUIT treatment. Data represent the mean ± standard errors. ^#^
*p* < 0.05, ^##^
*p* < 0.01 compared with level in the BISCUIT trial (Bonferroni test for post hoc); * *p* < 0.05, ** *p* < 0.01 compared with level in the BISCUIT trial (paired *t*-test). BISCUIT, regular snack; DF-BISCUIT, snacks with a rich dietary fiber content.

**Figure 4 foods-09-01500-f004:**
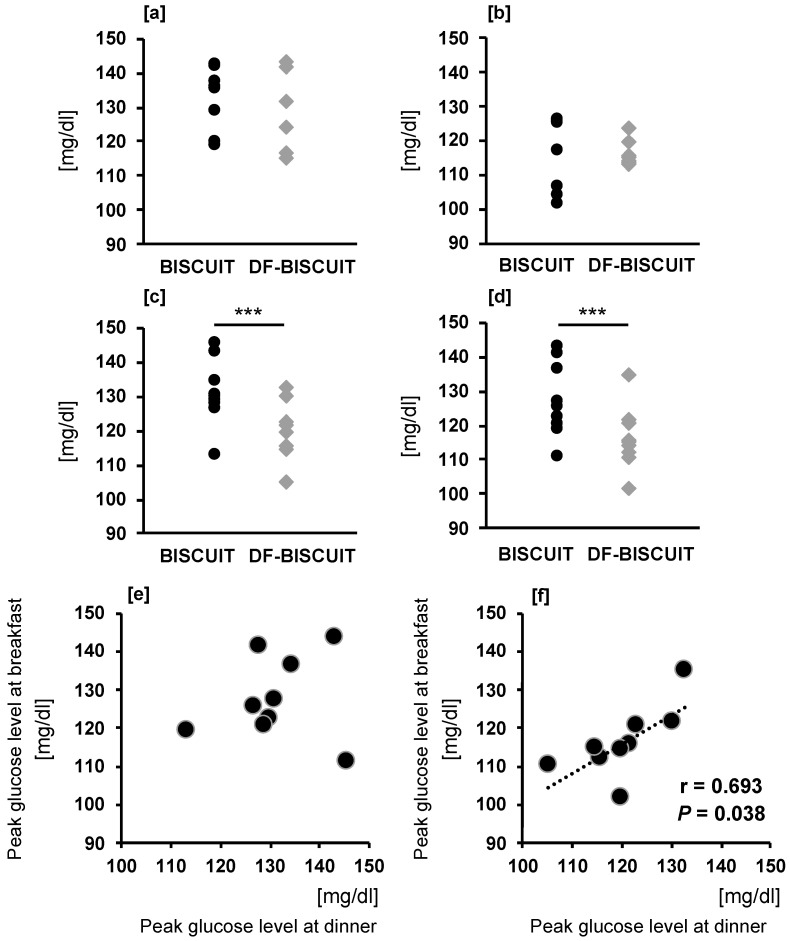
Peak glucose levels after lunch (**a**), after snack (**b**), after dinner (**c**), and after breakfast (**d**). *** *p* < 0.001 compared with level in the BISCUIT trial (paired *t*-test). The relationship between peak glucose level at breakfast and peak glucose level at dinner in the BISCUIT (**e**) trial and DF-BISCUIT (**f**) treatment. BISCUIT, regular snack; DF-BISCUIT, snacks with a rich dietary fiber content.

**Table 1 foods-09-01500-t001:** Snack contents, mealtimes, and physical activity levels in both trials.

	DF-BISCUIT	BISCUIT
Energy (kcal/day)	154	154
Protein (g/day)	2	1.7
Fat (g/day)	9.9	8.1
Total carbohydrate (g/day)	18.4	18.5
Sugar (g/day)	9.2	18.1
Dietary fiber (g/day)	9.2	0.4
Isomaltodextrin (g/day)	1.6	0
Inulin (g/day)	0.9	0
Cellulose (g/day)	0.3	0
Others (g/day)	6.4	0.4
Sodium chloride equivalent (g/day)	0.22	0.17
Mealtime		
Breakfast (h:min)	7:19 ± 0:13	7:18 ± 0:11
Lunch (h:min)	12:20 ± 0:32	12:19 ± 0:10
Snack (h:min)	15:32 ± 0:09	15:35 ± 0:08
Dinner (h:min)	18:28 ± 0:12	18:30 ± 0:15
Physical Activity		
MVPA (min/day)	86.0 ± 9.6	92.0 ± 12.2
Step counts (step/day)	8433.9 ± 1440.1	8260.9 ± 1100.1

Data on mealtime and physical activity levels represent the mean ± standard error. MVPA, moderate-to-vigorous physical activity. Others include processed resistant starch, flour, and macadamia.

**Table 2 foods-09-01500-t002:** Physical characteristic of all the participants at before and after the intervention.

	Before Intervention	After Intervention
Age (years)	76.9 ± 1.5	76.9 ± 1.6
Height (cm)	155.4 ± 2.7	155.4 ± 2.8
Body weight (kg)	52.9 ± 2.6	52.4 ± 2.5
BMI (kg/m^2^)	21.8 ± 0.6	21.1 ± 0.3
Energy intake (kcal/day)	2193.9 ± 288.9	2216.6 ± 255.8
Dietary fiber intake (g/day)	18.2 ± 2.1	16.9 ± 1.8

Data are mean ± standard error. BMI, body mass index.

**Table 3 foods-09-01500-t003:** Glucose parameters in both trials.

	DF-BISCUIT	BISCUIT	*p*
SD (mg/dL)	13.6 ± 0.6	15.8 ± 1.0	0.02 *
CV (%)	13.7 ± 0.6	15.4 ± 1.2	0.08
MAX (mg/dL)	130.1 ± 3.7	134.5 ± 2.3	0.20
MIN (mg/dL)	80.9 ± 3.1	80.4 ± 3.4	0.85
MAGE (mg/dL)	49.1 ± 1.6	54.1 ± 3.8	0.24

Data are mean ± standard error and were analyzed by a paired *t*-test. SD, standard deviation; CV, coefficient of variation; MAGE, mean amplitude of glycemic excursion; MAX, maximum glucose; MIN, minimum glucose. * *p* < 0.05 compared with level in the BISCUIT trial (paired *t*-test).

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
