# Peer review of "Effect of the Intake of a Snack Containing Dietary Fiber on Postprandial Glucose Levels"

_foods, 2020, doi:10.3390/foods9101500_

Round 1
Reviewer 1 Report
Postprandial glucose homeostasis is controlled by numerous factors. In this manuscript by Kim et al., the authors compared the blood glucose levels in healthy elderly people who consuming regular snack or a snack with a high dietary fiber content (SUNAO) between lunch and dinner for 1 week. Based on the results from a continuous glucose monitoring system, the SUNAO group showed a significant reduction of blood glucose levels and areas under the curve (AUCs). The manuscript is well organized and concise. Here are some comments.
- Since, among participants, there are 4 men and 5 women, did the author observe any differences between different sex?
- There is 9.1 g/day sugar in the SUNAO snack; however, there is 18.1 g/day sugar in Biscuit snack. Does this different amount of sugar have an effect on blood glucose levels after consuming for one week?
Author Response
Paper No. foods-915265
Response to the Reviewers’ comments
Once again, we wish to thank the editor and reviewers for reading our manuscript so thoroughly and providing such constructive feedback. The quality of our manuscript has certainly improved as a result of these comments. Our responses are provided below in a point-by-point fashion and any changes in the revised manuscript are highlighted in yellow.
Reviewer 1:
Postprandial glucose homeostasis is controlled by numerous factors. In this manuscript by Kim et al., the authors compared the blood glucose levels in healthy elderly people who consuming regular snack or a snack with a high dietary fiber content (SUNAO) between lunch and dinner for 1 week. Based on the results from a continuous glucose monitoring system, the SUNAO group showed a significant reduction of blood glucose levels and areas under the curve (AUCs). The manuscript is well organized and concise. Here are some comments.
Query 1:
Since, among participants, there are 4 men and 5 women, did the author observe any differences between different sex?
Response 1:
Thank you for your comment. In this study, we compared the effects of snack intake on blood glucose fluctuations between sexes. Males showed similar variations to the overall results, but females did show small differences between the DF-BISCUIT treatment and the BISCUIT trial. Therefore, the effect of snack consumption rich in dietary fiber on blood glucose fluctuations may differ depending on the sex. However, due to the small number of participants, statistical processing is not performed. We have added this issue to the Results (lines 173–178).
Query 2:
There is 9.1 g/day sugar in the SUNAO snack; however, there is 18.1 g/day sugar in Biscuit snack. Does this different amount of sugar have an effect on blood glucose levels after consuming for one week?
Response 2:
Thank you for your comment. In this study, the total energy content of the snacks in both trials is similar. Therefore, it is considered that the influence of both trials on the blood glucose level fluctuation is largely due to the difference in dietary fiber contained in the snack. However, the effects of different types and amounts of sugar cannot be denied, so it is necessary to consider these factors in the future. We have added this issue to the Discussion (lines 251–256).
Once again, we thank the editor and reviewers for reading our manuscript so thoroughly and providing such constructive feedback. We hope that the modifications made in the revised version and our responses to the reviewers’ concerns will be sufficient to render the paper suitable for publication in Foods.

Reviewer 2 Report
The authors have conducted a human trial to test the effect of the intake of snacks with high fiber content in between lunch and dinner on glucose homeostasis. It is a new angle to determine the effect of fiber benefits on glucose metabolism and control. The observations are interesting and helpful for the health promotion and maintenance of the general population and those who have family history of hyperglycemia and other risk factors for developing insulin resistance and diabetes. The manuscript is well written but a revision is needed to address a few comments/questions listed below.
- The authors should provide the number of participants at the beginning of treatment and the number who completed the trial, and also the number of participants who were excluded from the final data analysis due to failure to meet the inclusion criteria.
- What are the ratios among the three different types of fiber? This information should be linked to the discussion of each type of fiber on glucose metabolism and control.
- The participants were asked to wear a triaxial accelerometer every day at all times in the experimental design. However, the data were considered valid if the participants wore it for a minimum of 10 hours a day for at least 2 weekdays and a weekend day. Any reasons for such as discrepancy?
- Figure 1, there was no washout in between the two treatments. The carry over effect could have masked part of the second treatment on the same subject. Please discuss it. Also, in figure 1, the time may be labeled as week 1 and week 2 instead 1 week and 2 week.
- In the text, the SUNAO trail should be changed to SUNAO treatment.
- The authors should provide information regarding food and energy intake after the addition of snacks as compared with data of the baseline or before treatments. This would be helpful if the intake of snacks affect (increase/decrease) energy balance and the body weight in a long run.
- The sample size was small and should be discussed.
Author Response
Paper No. foods-915265
Response to the Reviewers’ comments
Once again, we wish to thank the editor and reviewers for reading our manuscript so thoroughly and providing such constructive feedback. The quality of our manuscript has certainly improved as a result of these comments. Our responses are provided below in a point-by-point fashion and any changes are highlighted in yellow in the revised manuscript.
Reviewer 2:
The authors have conducted a human trial to test the effect of the intake of snacks with high fiber content in between lunch and dinner on glucose homeostasis. It is a new angle to determine the effect of fiber benefits on glucose metabolism and control. The observations are interesting and helpful for the health promotion and maintenance of the general population and those who have family history of hyperglycemia and other risk factors for developing insulin resistance and diabetes. The manuscript is well written but a revision is needed to address a few comments/questions listed below.
Query 1:
The authors should provide the number of participants at the beginning of treatment and the number who completed the trial, and also the number of participants who were excluded from the final data analysis due to failure to meet the inclusion criteria.
Response 1:
As you suggested, we have rewritten this sentence to reflect this information (lines 78–79).
Query 2:
What are the ratios among the three different types of fiber? This information should be linked to the discussion of each type of fiber on glucose metabolism and control.
Response 2:
As you suggested, we have added the data for the three different types of fiber (Table 2, and lines 152–155).
Query 3:
The participants were asked to wear a triaxial accelerometer every day at all times in the experimental design. However, the data were considered valid if the participants wore it for a minimum of 10 hours a day for at least 2 weekdays and a weekend day. Any reasons for such as discrepancy?
Response 3:
Thank you for your comment. A previous study reported that at least two weekdays and a weekend day were required to evaluate the amount of physical activity per day using a triaxial accelerometer 1). Therefore, we have used the same criteria in this study. As you indicated, using all of the data is more effective for the evaluation of the daily physical activity. However, some participants forgot to wear an accelerometer for enough time and some wore it improperly. Thus, we have evaluated daily physical activity based on the above criteria.
1) Mâsse LC et al., Med Sci Sports Exerc. 2005, 37(11 Suppl):S544-54. doi: 10.1249/01.mss.0000185674.09066.8a.
Query 4:
Figure 1, there was no washout in between the two treatments. The carry over effect could have masked part of the second treatment on the same subject. Please discuss it.
Response 4:
Thank you for your comment. We have added this issue to the limitations (lines 265–267). The carryover effect in this study cannot be denied. Therefore, to minimize the carryover effect, the number of participants who started each trial was divided equally as much as possible. Thus, the carryover effect between both trials is considered to be small.
Query 5:
Also, in figure 1, the time may be labeled as week 1 and week 2 instead 1 week and 2 week.
Response 5:
Thank you for your suggestion. We have reworded “1 week and 2 week” to “week 1 and week 2” (lines 97–99).
Query 6:
In the text, the SUNAO trial should be changed to SUNAO treatment.
Response 6:
Based on your suggestion, we have reworded “SUNAO trial” to “DF-BISCUIT treatment.”
Query 7:
The authors should provide information regarding food and energy intake after the addition of snacks as compared with data of the baseline or before treatments. This would be helpful if the intake of snacks affect (increase/decrease) energy balance and the body weight in a long run.
Response 7:
Thank you for your comment. We sincerely apologize for having caused any confusion. We used the Food Frequency Questionnaire (FFQ) to evaluate their energy intake at baseline and after intervention (Table 1). The baseline assessment represents energy intake before intervention. On the other hand, FFQ evaluation after the intervention is reflected as the energy intake during and after the intervention. Participants were also asked to maintain their dietary habits throughout the two weeks of the experiment. As a result, there was no statistically significant difference in energy intake between baseline and following the intervention. Therefore, eating snacks of at least less than 200 kcal for 1 week is considered to have little impact on body composition.
Query 8:
The sample size was small and should be discussed.
Response 8:
Thank you for your suggestion. We have added this point to the limitations (lines 267–270).
Once again, we thank the editor and reviewers for reading our manuscript so thoroughly and providing such constructive feedback. Hopefully, the new version of the manuscript and our responses to the reviewers’ concerns will be sufficient to render the paper suitable for publication in Foods.

Reviewer 3 Report
The purpose of the current study was to examine the effect of the intake of snack that contain contains dietary fiber under free-living conditions on post-prandial glucose levels excursions. The paper is well written and addresses a relevant issue for the journal. However, some points need to be addressed.
Points:
L55: Rather than writing “reduced glucose digestion” it is more correct to write ”prolonged starch digestion” – it is not glucose that is digested but starch that is digested and converted into glucose that is absorbed.
L63-64: What is the hypothesis to be tested?
L77-78: The information concerning the types of dietary fiber used in the SUNAO snack (L210-212) should be presented here rather than in the Discussion section.
L86: Can the snack be taken at any time between lunch and dinner when hungry or is there any recommended time?
L112-113 & Table 1: It is unclear for me if the daily intake referred to in Table 1 is the daily intake the subjects are consuming after they have been through the interventions; it looks so in Figure 1. However, what is the intake of dietary fiber then with the two snacks during the intervention?
L142 - Table 2: Is it possible that the lower sugar content present in the SUNAO snack can explain the difference in plasma glucose between the two interventions?
L145 – Table 3: It is unclear to me how SD and CV’s that both represent variabilities in itself can have variations? Please explain.
L216-220: An effect via the metabolism in the large intestine usually requires adaptation. Will one-week duration, as used in this study of the intervention, be sufficient?
L232-240: A limitation that also should be mentioned and discussed is the number of participants, which seems to be in the lower end.
Author Response
Paper No. foods-915265
Response to the Reviewers’ comments
Once again, we wish to thank the editor and reviewers for reading our manuscript so thoroughly and providing such constructive feedback. The quality of our manuscript has certainly improved as a result of these comments. Our responses are provided below in a point-by-point fashion and any changes are highlighted in yellow in the revised manuscript.
Reviewer 3:
The purpose of the current study was to examine the effect of the intake of snack that contain contains dietary fiber under free-living conditions on post-prandial glucose levels excursions. The paper is well written and addresses a relevant issue for the journal. However, some points need to be addressed.
Query 1:
L55: Rather than writing “reduced glucose digestion” it is more correct to write ”prolonged starch digestion” – it is not glucose that is digested but starch that is digested and converted into glucose that is absorbed.
Response 1:
Based on your suggestion, we have reworded “reduced glucose digestion” to “prolonged starch digestion” (line 56).
Query 2:
L63-64: What is the hypothesis to be tested?
Response 2:
Thank you for your comments. We added the hypothesis of this research to the revised version (lines 65–67).
Query 3:
L77-78: The information concerning the types of dietary fiber used in the SUNAO snack (L210-212) should be presented here rather than in the Discussion section.
Response 3:
Thank you for your suggestion. We added the types of dietary fiber used in the SUNAO snack of this research to Methods section (Table 2, and lines 152–154).
Query 4:
L86: Can the snack be taken at any time between lunch and dinner when hungry or is there any recommended time?
Response 4:
We apologize for the lack of explanation regarding the snack intake pattern. We have rewritten this sentence to reflect this information (line 89). In this study, the snack intake time was instructed to be taken in the middle of lunch and dinner, not at any time between lunch and dinner.
Query 5:
L112-113 & Table 1: It is unclear for me if the daily intake referred to in Table 1 is the daily intake the subjects are consuming after they have been through the interventions; it looks so in Figure 1. However, what is the intake of dietary fiber then with the two snacks during the intervention?
Response 5:
Thank you for your comment. FFQ is a questionnaire used to obtain frequency and, in some cases, portion size information about food consumption over a specified period of time, typically for the past one month. Dietary fiber intake during each intervention in the DF-BISCUIT and BISCUIT trials is unclear. However, since the effect of daily energy intake on blood glucose fluctuation cannot be ruled out, FFQ was used to evaluate energy intake and dietary fiber intake before and after the experiment. As a result, as shown in Table 1, there was no difference in dietary fiber intake and energy intake between the two trials before and after the experiment. Furthermore, participants were also asked to maintain their dietary habits throughout the two weeks of the experiment. Therefore, it is considered that there is no significant difference in dietary fiber intake between trials during the intervention period.
Query 6:
L142 - Table 2: Is it possible that the lower sugar content present in the SUNAO snack can explain the difference in plasma glucose between the two interventions?
Response 6:
Thank you for your comment. In this study, the total energy content of the snacks in both trials is similar. Therefore, it is considered that the influence of both trials on the blood glucose level fluctuation is largely due to the difference in dietary fiber contained in the snack. However, the effects of different types and amounts of sugar cannot be denied, so it is necessary to consider these factors in the future. We have added this issue to the Discussion (lines 251–256).
Query 7:
L145 – Table 3: It is unclear to me how SD and CV’s that both represent variabilities in itself can have variations? Please explain.
Response 7:
SD (standard deviation) and CV (coefficient of variation) are two statistical values that provide a different lens to look at glucose variability. These parameters are often used in previous studies 1,2,3). SD is a measure of the spread in glucose readings around the average. For example, if someone's blood glucose level fluctuates between high and low levels one day, SD will increase. On the other hand, those who have a stable day with little blood glucose fluctuation will have lower SD. However, SD needs to be interpreted in consideration of the mean blood glucose level. Therefore, the CV will be used. SD is strongly influenced by the mean glucose level, while CV helps to “correct” and normalize glucose variability. Therefore, SD and CV should be as low as possible to maintain stable blood glucose levels. We have added this information to the Methods section (lines 108–112).
1) Imai S et al., Diabetes Metab. 2018, 44(6): 482-487. doi: 10.1016/j.diabet.2018.07.001.
2) Nitta A et al., Diabetes Metab. 2019, 45(4): 369-374. doi: 10.1016/j.diabet.2018.10.004.
3) Monnier L et al., Diabetes Care. 2017, 40(7):832-838. doi: 10.2337/dc16-1769.
Query 8:
L216-220: An effect via the metabolism in the large intestine usually requires adaptation. Will one-week duration, as used in this study of the intervention, be sufficient?
Response 8:
Thank you for your comment. A previous study has shown that a 7-day inulin intake changes the intestinal environment in humans 1). Therefore, even a week’s intake may be sufficiently adapted. We have added this issue to the Discussion (lines 235–236).
1)Bouhnik Y et al., Nutr J. 2006, 5:8. doi: 10.1186/1475-2891-5-8.
Query 9:
L232-240: A limitation that also should be mentioned and discussed is the number of participants, which seems to be in the lower end.
Response 9:
Thank you for your suggestion. We have added this to the limitations (lines 267–270).
Once again, we thank the editor and reviewers for reading our manuscript so thoroughly and providing such constructive feedback. Hopefully, the new version of the manuscript and our responses to the reviewers’ concerns will be sufficient to render the paper suitable for publication in Foods.

Round 2
Reviewer 3 Report
The authors have responded to the queries in a proper way and I can now recommend acceptance of the manuscript.